# Inhibitory Effects of a Novel μ-Opioid Receptor Nonpeptide Antagonist, UD-030, on Morphine-Induced Conditioned Place Preference

**DOI:** 10.3390/ijms24043351

**Published:** 2023-02-08

**Authors:** Soichiro Ide, Noriaki Iwase, Kenichi Arai, Masahiro Kojima, Shigeru Ushiyama, Kaori Taniko, Kazutaka Ikeda

**Affiliations:** 1Addictive Substance Project, Tokyo Metropolitan Institute of Medical Science, Tokyo 156-8506, Japan; 2UBE Corporation, Yamaguchi 755-8633, Japan; 3Sanwa Kagaku Kenkyusho Co., Ltd., Aichi 461-8631, Japan

**Keywords:** opioid receptor antagonist, morphine, addiction, CPP, reward, naltrexone

## Abstract

Although opioids are widely used to treat moderate to severe pain, opioid addiction and the opioid overdose epidemic are becoming more serious. Although opioid receptor antagonists/partial agonists, such as naltrexone and buprenorphine, have relatively low selectivity for the μ-opioid receptor (MOP), they have been used for the management of opioid use disorder. The utility of highly selective MOP antagonists remains to be evaluated. Here, we biologically and pharmacologically evaluated a novel nonpeptide ligand, UD-030, as a selective MOP antagonist. UD-030 had more than 100-fold higher binding affinity for the human MOP (K_i_ = 3.1 nM) than for δ-opioid, κ-opioid, and nociceptin receptors (K_i_ = 1800, 460, and 1800 nM, respectively) in competitive binding assays. The [^35^S]-GTPγS binding assay showed that UD-030 acts as a selective MOP full antagonist. The oral administration of UD-030 dose-dependently suppressed the acquisition and expression of morphine-induced conditioned place preference in C57BL/6J mice, and its effects were comparable to naltrexone. These results indicate the UD-030 may be a new candidate for the treatment of opioid use disorder, with characteristics that differ from traditional medications that are in clinical use.

## 1. Introduction

Opioids are widely used to treat moderate to severe pain that is insufficiently controlled by other medications. However, the regular nonmedical use of opioids, prolonged use, misuse, and use without medical supervision can lead to opioid addiction and other health problems. Opioid addiction and the opioid overdose epidemic (i.e., the “opioid crisis”) have become increasingly more serious worldwide, although there are countries where prescriptions are low and adequate pain management is not being provided [1]. Opioids exert their pharmacological effects by mainly acting on μ (MOP), κ (KOP), and δ (DOP) opioid receptors [2]. Some opioids, such as buprenorphine, are also reported to act on the nociceptin/orphanin FQ receptor (NOP/ORL-1) [3,4]. Studies of inbred and gene knockout mice have clearly shown that the MOP plays a major role in the analgesic and rewarding effects of opioids [5,6], and the KOP is also involved in their analgesic effects on visceral pain [7,8]. Thus, MOP antagonists are essential for studying the involvement of MOPs in drug abuse and the efficacy of medications for addiction. Opioid receptor antagonists have been developed to counteract the adverse effects of opioids. Naloxone and naltrexone (NTX), which are the two most commonly used opioid receptor antagonists, are used primarily for the treatment of opioid overdose and alcohol and opioid dependence, respectively. Buprenorphine, an opioid receptor partial agonist [9], has also been used to treat opioid use disorder and acute and chronic pain [10]. Although these antagonists/partial agonists with relatively low selectivity for the MOP have been shown to inhibit relapse and curb drug cravings, the utility of highly selective MOP antagonists remains to be evaluated.

Some moderately potent ligands are available, but few optimal nonpeptide antagonists have been developed for the MOP. Most highly selective MOP antagonists that are currently available are conformation-constrained peptides, such as CTOP (D-Phe-Cys-Tyr-D-Trp-Orn-Thr-Pen-Thr-NH_2_) and CTAP (D-Phe-Cys-Tyr-D-Trp-Arg-Thr-Pen-Thr-NH_2_). They have been used to target the MOP in in vitro and in vivo studies, but their limited bioavailability may not be suitable for many types of in vivo studies and medical applications [11,12,13]. Peptides generally cannot be administered orally because they are quickly degraded by digestive enzymes in the gastrointestinal tract. Additionally, in many cases, long-chain peptides have low membrane permeability because of the high number of water-soluble functional groups, resulting in poor pharmacokinetics in the body. Although peptide preparations that mitigate these disadvantages have been developed, there have been few successful examples of their medical use. Thus, nonpeptide selective MOP antagonists could be potential oral therapeutic agents for addiction. The aim of the present study was to determine the binding selectivity of UD-030, a novel nonpeptide ligand, for each opioid receptor subtype and its agonist/antagonist activity. Conditioned place preference (CPP) tests were conducted to test the inhibitory effect of UD-030 on morphine preference and compare it to the clinically used NTX.

## 2. Results

### 2.1. In Vitro Assays of UD-030

UD-030 (Figure 1A) was first evaluated in a competitive radioligand binding assay using monocloned opioid receptors that were expressed in Chinese hamster ovary (CHO) cell lines. [^3^H]DAMGO, [^3^H]DADLE, [^3^H]U69593, and [^3^H]nociceptin were used to label the MOP, DOP, KOP, and NOP, respectively. The binding affinity of UD-030 for each receptor is shown in Figure 1. The calculated K_i_ values are shown in Table 1. UD-030 showed nanomolar affinity (K_i_ = 3.1 nM) for the MOP and over 100-fold higher selectivity for the MOP over the DOP, KOP, and NOP. Naltrexone has been reported to have subnanomolar-to-nanomolar affinity for the three opioid receptors and shows mild selectivity for the MOP and KOP over the DOP (Table 1). Naltrexone is reported to have low selectivity for the MOP over the KOP [14]. These results indicate that UD-030 is a highly MOP-selective nonpeptide opioid ligand.

To evaluate the agonistic and antagonistic actions of UD-030 on various opioid receptors, UD-030 was tested in the [^35^S]-GTPγS binding assay (Table 1). Although all reference compounds (competitive agonists; DAMGO for the MOP, DPDPE for the DOP, U-69593 for the KOP, and nociceptin for the NOP) showed potent agonistic activity (maximal effect > 95% for all tested receptors; EC_50_ values are shown in Table 1) on each opioid receptor that was expressed in cells, UD-030 did not stimulate [^35^S]-GTPγS binding even when applied at a high concentration (10 µM). The assay of antagonistic activity showed that UD-030 completely and concentration-dependently inhibited the effect of each reference compound (maximal inhibitory effect > 95% for all tested receptors; IC_50_ values are shown in Table 1). These results indicated that UD-030 had no agonistic activity on any opioid receptors but had very strong antagonistic activity on the MOP.

### 2.2. Pharmacokinetics of Orally Administered UD-030

To investigate the potential usefulness of the UD-030 nonpeptide ligand as an oral agent, we conducted a preliminary validation in a small number of mice to determine whether it is absorbed after oral administration and predict the peak and duration of its effects after administration. The concentration of UD-030 in plasma was measured after oral administration once in male mice at 0.5, 1, 2, and 4 mg/kg. The time course of the plasma concentrations of UD-030 is shown in Figure 2. Plasma concentrations of UD-030 and pharmacokinetic parameters are shown in Table 2. After oral administration, an increase in plasma UD-030 concentration was observed. The time to reach the maximum plasma concentration (T_max_) was 1.0–8.0 h, it became the lower limit of detection in plasma at 24 h. The maximum plasma concentration (C_max_) was 4.95, 14.9, 36.1, and 119 ng/mL for the 0.5, 1, 2, and 4 mg/kg doses. The area under the plasma concentration-time curve (AUC_0–24h_) was 72.3, 176, 350, and 1220 ng·h/mL, respectively. Both C_max_ and AUC_0–24h_ dose-dependent increased. These results indicated that oral UD-030 administration is well absorbed and has relatively long-lasting effects. The schedule of the behavioral experiments was then determined based on these results.

### 2.3. Inhibitory Effect of UD-030 on the Acquisition of Morphine-Induced Conditioned Place Preference

Because of its highly selective antagonistic activity against the MOP, the inhibitory effect of UD-030 on the rewarding effects of morphine was compared with NTX, a nonselective opioid receptor antagonist. We assessed rewarding effects using the CPP test. Mice were pretreated with UD-030 (0, 3, and 10 mg/kg, oral [p.o.]) or NTX (10 mg/kg, p.o.) 60 min before each morphine treatment during the conditioning phase. The two-way repeated measures analysis of variance (ANOVA) showed significant differences between the time spent in the pre- and postconditioning phases (pre vs. post, *F*_1,28_ = 4.78, *p* = 0.037: interaction, *F*_3,28_ = 9.05, *p* < 0.001). Morphine significantly increased the time spent in the previously paired compartment in vehicle-pretreated control mice (preconditioning: 443.1 ± 33.2 s; postconditioning: 554.4 ± 31.8 s; *p* < 0.001, Sidak’s multiple-comparison test; Figure 3A). No significant differences were found between the time spent in the pre- and postconditioning phases in drug-pretreated groups (3 mg/kg UD-030, preconditioning: 445.2 ± 18.9 s, postconditioning: 470.6 ± 16.5 s; 10 mg/kg UD-030, preconditioning: 434.9 ± 30.5 s, postconditioning: 392.8 ± 42.3 s; 10 mg/kg NTX, preconditioning: 437.0 ± 38.1 s, postconditioning: 436.5 ± 49.7 s; Figure 3A). The ANOVA of CPP scores showed significant differences among the pretreated drugs before morphine treatment (*F*_3,28_ = 9.05, *p* < 0.001). *Post hoc* comparisons indicated that morphine-induced CPP scores were significantly lower in the drug-pretreated groups (3 mg/kg UD-030, *p* = 0.023; 10 mg/kg UD-030, *p* < 0.001; 10 mg/kg NTX, *p* = 0.003; Dunnett’s multiple-comparison *post hoc* test; Figure 3B) compared with the vehicle-pretreated group.

### 2.4. Inhibitory Effect of UD-030 on the Expression of Morphine-Induced CPP

To assess the inhibitory effect of UD-030 on the expression of morphine-induced rewarding effects, we conducted the morphine-induced CPP test. Mice were treated with UD-030 or NTX 60 min before the postconditioning phase. The two-way repeated-measures ANOVA showed significant differences between the time spent in the pre- and postconditioning phases (pre vs. post, *F*_1,28_ = 31.95, *p* < 0.0001: interaction, *F*_3,28_ = 3.05, *p* = 0.045). Morphine significantly increased the time spent in the previously paired compartment in the vehicle-treated control mice (preconditioning: 443.6 ± 34.4 s; postconditioning: 554.4 ± 29.7 s; *p* < 0.001, Sidak’s multiple-comparison test; Figure 4A) and in 3 mg/kg UD-030-treated mice (preconditioning: 466.8 ± 20.7 s; postconditioning: 536.5 ± 24.8 s; *p* = 0.012, Sidak’s multiple-comparison test; Figure 4A). There were no significant differences between the time spent in the pre- and postconditioning phases in the other groups (10 mg/kg UD-030, preconditioning: 461.3 ± 20.0 s, postconditioning: 492.4 ± 32.8 s; 10 mg/kg NTX, preconditioning: 441.6 ± 33.0 s, postconditioning: 474.1 ± 29.9 s; Figure 4A). The one-way ANOVA of CPP scores showed significant differences among the treated drugs before the preconditioning session (*F*_3,28_ = 3.05, *p* = 0.045). *Post hoc* comparisons indicated that morphine-induced CPP scores significantly decreased in 10 mg/kg UD-030- and 10 mg/kg NTX-treated mice (*p* = 0.038 and *p* = 0.042; Dunnett’s multiple-comparison *post hoc* test; Figure 4B) compared with the vehicle-treated control group.

In the postconditioning phase, there were no significant differences in the number of transitions between compartments among groups (one-way ANOVA: *F*_3,28_ = 0.82, *p* = 0.49), indicating that 10 mg/kg UD-030 and NTX did not exert sedative effects at this dose. No other notable behavioral abnormalities were observed in mice after high-dose (10 mg/kg, p.o.) UD-030 treatment during the session.

## 3. Discussion

In the present study, we found that the novel selective MOP nonpeptide antagonist UD-030 may have the potential to be a seed compound for the treatment of opioid use disorder. UD-030 induced dose-dependent inhibitory effects on morphine-induced CPP when administered orally. The inhibitory effect of 10 mg/kg of UD-030 on morphine-induced CPP was equivalent to the effect of 10 mg/kg NTX. Interestingly, both UD-030 and NTX significantly suppressed both the acquisition and expression of morphine-induced CPP. UD-030 may exert its inhibitory effects on the formation of rewarding effects of morphine by antagonistically inhibiting binding to the MOP. Not only NTX but also other opioid receptor antagonists, such as naloxone, have been reported to cause conditioned place aversion (CPA) [15,16]. No conditioning with UD-030 treatment alone was performed in the present study, which is a limitation. However, a slight, although nonsignificant, tendency toward CPA was observed in the 10 mg/kg UD-030 and morphine conditioning group, suggesting that UD-030, like other opioid antagonists, could produce CPA. Nonetheless, UD-030 inhibited the expression of morphine-induced CPP after acquisition, indicating that UD-030-induced CPA may not simply counteract morphine-induced CPP but may suppress rewarding effects of morphine via the modulation of endogenous opioid neural activity. These findings support our hypothesis that selective MOP antagonists are effective for blocking rewarding effects of opioids, at least for morphine. Naltrexone has been used for the management of alcohol dependence [17]. It has also been reported to inhibit dependence on the psychostimulant methamphetamine, which has a different mechanism of action from opioids, and is being studied for clinical use [18,19]. Genetic polymorphisms of opioid receptor genes have been reported to be associated with addiction to opioids and other addictive substances [20,21,22]. These findings suggest that actions on the endogenous opioid system may be a common mechanism of substance use disorders. UD-030 may also be effective for other substance use disorders, such as alcohol and psychostimulants, but further studies are needed.

The two most commonly used centrally acting opioid receptor antagonists are naloxone and NTX. However, both naloxone and NTX have low selectivity for the MOP and may cause adverse side effects because of this low selectivity [23,24]. Methylnaltrexone is the most commonly used peripheral opioid receptor antagonist. Methylnaltrexone is a potent competitive antagonist that acts on the digestive tract and is used for the treatment of opioid-induced constipation [25,26]. Nalmefene is another opioid antagonist that is used for the management of opioid overdose and alcohol dependence [27]. All of these clinically used opioid receptor antagonists have low selectivity between the MOP and KOP. Other types of MOP-selective antagonists have been developed, including peptide antagonists (e.g., CTAP and CTOP) and the irreversible nonpeptide ligand β-FNA [28]. These peptides and irreversible nonpeptide compounds are not used for medicinal purposes because of problems such as rapid metabolism and permanent receptor inactivation. In the present study, the novel UD-030 nonpeptide ligand showed high affinity for the MOP and higher selectivity for the MOP compared with NTX. This may indicate that UD-030 has the potential to be a centrally transferable drug with fewer side effects that are caused by low subtype selectivity. The development of reversible and selective nonpeptide MOP antagonists, such as UD-030, could be beneficial for the treatment of adverse effects of opioid receptor agonists (e.g., morphine). The concept of “functional selectivity” or “biased agonism” has been attracting attention in recent years with regard to separating analgesic effects of MOP agonists from their adverse effects [29]. The activation of MOPs by agonists triggers various processes via downstream signaling through G proteins and β-arrestins. Although most MOP agonists that have been reported to date are non-biased ligands, G-protein signaling pathway-biased MOP ligands have been hypothesized to provide analgesia with fewer adverse effects [29]. Antagonistic effects of UD-030 on these biased and non-biased ligands will need to be tested in the future.

Most opioid analgesics in current clinical use are known to have binding affinity for the MOP and KOP, with the exception of fentanyl and its derivatives, which have very high selectivity for the MOP. Stimulation of the MOP produces clinically relevant analgesia, but it also produces respiratory depression and dependence. Stimulation of the KOP also exerts analgesic effects, but it has been reported to cause disassociation, hallucinations, and dysphoria [30]. Our previous studies with inbred and MOP gene knockout mice have also shown that the MOP plays a major role in exerting analgesic and rewarding effects of opioids [5,6], and the KOP is also partially involved in their analgesic effects on visceral pain [7,8]. Thus, opioid analgesics are thought to exert their analgesic effects primarily through the MOP and partially through the KOP, but they produce dependence primarily through the MOP. At present, it is not possible to separate the analgesic effect from dependence via MOPs. Although not studied here, UD-030, like other opioid receptor antagonists with binding affinity for MOP, could be thought to inhibit analgesic effects of opioids. While the problem of opioid use disorder continues to grow, the fact that patients with opioid use disorder still suffer from chronic pain is also a major problem from a treatment perspective [31,32]. It is unclear whether opioid receptor agonist treatment should be maintained or tapered for patients with both chronic pain and opioid use disorder. For these patients, methadone or a buprenorphine/naloxone fixed-dose combination are currently prescribed as opioid replacement therapy. Fixed-dose combinations of MOP-selective antagonists and opioid receptor agonists may lead to new therapies that partially suppress MOPs while maintaining analgesic effects of KOP activation. Future studies should determine specific ratios of combinations that can maintain analgesic effects while reducing dependence potential such that selective MOP antagonists are developed for the treatment of chronic pain and comorbid addiction with fewer side effects.

One of the main limitations of the present study, in addition to not performing conditioning with UD-030 alone, was the lack of assurance of the reliability and reproducibility of pharmacokinetic studies. The present study was conducted by selecting a general time course and method for a preliminary positioning study to confirm the approximate timing of peak plasma concentration and half-life before conducting the behavioral experiments for the purpose of developing the drug for oral administration. Future validation with matching animal species and doses and comparisons with other routes of administration will be required. Another limitation was the lack of reward-related experiments beyond CPP, such as self-administration and intracranial self-stimulation. The CPP test can be influenced by memory. Other reward-related studies should be conducted in the future. Moreover, further studies of UD-030 and its derivatives are required to verify various other physiological effects that are applicable to human studies. Putative agents, such as MOP antagonists, that reduce the nonmedical use of prescription opioids may also negatively impact patient compliance if aversive effects of the putative agent are too strong. Another consideration is the complementarity of pharmacokinetic profiles of paired drugs in a formulation.

In conclusion, the novel nonpeptide ligand UD-030 is a highly selective antagonist of the MOP over the DOP, KOP, and NOP. The oral administration of UD-030 dose-dependently suppressed the acquisition and expression of the rewarding effects of morphine. UD-030 may be a new candidate for the treatment of opioid use disorder, with characteristics that differ from traditional medications that are in clinical use.

## 4. Materials and Methods

### 4.1. Drugs

For the in vitro assays, salt-free UD-030 (gift from UBE Corporation, Yamaguchi, Japan), DAMGO (MOP-selective agonist), and U-69593 (a KOP-selective agonist; Sigma Aldrich, St. Louis, MO, USA) were dissolved in dimethylsulfoxide. DADLE, DPDPE (DOP agonists; Sigma Aldrich), and nociceptin (NOP endogenous agonist; Enzo Life Sciences, Farmingdale, NY, USA) were dissolved in each assay buffer that is described below.

For the pharmacokinetic and pharmacological studies, morphine hydrochloride (Takeda Pharma, Osaka, Japan) was dissolved and diluted in saline and administered intraperitoneally (i.p.) in a volume of 10 mL/kg body weight. UD-030 and naltrexone hydrochloride (Tocris Bioscience, Bristol, UK) were dissolved and diluted in sterilized 0.5 *w*/*v*% Methyl Cellulose 400 Solution (FUJIFILM Wako Pure Chemical Co., Osaka, Japan) and administered orally (p.o.) in a volume of 10 mL/kg body weight. All experiments were conducted with a dose of morphine (10 mg/kg) that is sufficient to elicit CPP.

### 4.2. Radioligand Binding Assay

Chinese hamster ovary cell lines that stably expressed the human MOP, DOP, KOP, and NOP (hMOP/CHO, hDOP/CHO, hKOP/CHO, and hNOP/CHO, respectively) were obtained from Cosmo Bio Co., Ltd. (Tokyo, Japan). Expressing cells were harvested and homogenized in ice-cold assay buffer (50 mM Tris-HCl and 5 mM MgCl_2_, pH 7.4) twice with a Polytron homogenizer at 12,000 rotations per minute (rpm) for 20 s. These solutions were then diluted with assay buffer to obtain the final concentration (MOP: 18.2 µg/mL, DOP: 18.2 µg/mL, KOP: 63.6 µg/mL, and NOP: 81.8 µg/mL, respectively). For the competitive binding assays, these cell membrane suspensions were stirred at 1100 rpm for 60 min on a plate shaker at room temperature with [^3^H]DAMGO for hMOP (final concentration, 1 nM), [^3^H]DADLE for hDOP (final concentration, 0.7 nM), [^3^H]U-69593 for hKOP (final concentration, 1 nM), or [^3^H]nociceptin for hNOP (final concentration, 0.5 nM) in the presence of a WGA SPA Beads (PerkinElmer, Waltham, MA, USA) suspension and various concentrations of ligands. Nonspecific binding was determined in the presence of unlabeled 100 μM DAMGO for hMOP, 50 μM DADLE for hDOP, 100 μM U69593 for hKOP, or 50 μM nociceptin for hNOP. After stirring for 60 min and being centrifuged at 1000 rpm for 3 min at room temperature, radioactivity was measured by MicroBeta^2^ (PerkinElmer). K_i_ values were calculated by curve fitting using Prism 7 software (GraphPad, San Diego, CA, USA) with the nonlinear regression–one-site binding model using the K_d_ value of each type of receptor (hMOP: K_d_ = 2.498 nM, hDOP: K_d_ = 0.673 nM, hKOP: K_d_ = 1.107 nM, and hNOP: K_d_ = 0.150 nM). Assays were performed on two separate occasions, each in triplicate.

### 4.3. [^35^S]-GTPγS Binding

hMOP/CHO, hDOP/CHO, hKOP/CHO, and hNOP/CHO were homogenized in ice-cold GTP assay buffer (100 mM NaCl, 5 mM MgCl_2_, 1 mL EDTA, and 50 mM HEPES, pH 7.4) twice with a Polytron homogenizer at 12,000 rpm for 20 s. These solutions were then diluted with GTP assay buffer that contained 18.2 μM guanosine diphosphate (final) to obtain the final protein concentration (MOP: 36.0 µg/mL, DOP: 27.0 µg/mL, KOP: 64.0 µg/mL, and NOP: 64.0 µg/mL, respectively). [^35^S]GTP was diluted with GTP assay buffer to a final concentration of 0.08 nM. These cell membrane suspensions were then stirred at 1100 rpm for 60 min on the plate shaker at 30 °C with [^35^S]GTP (final concentration, 0.08 nM) in the presence of the WGA SPA Beads suspension and various concentrations of ligands. After stirring for 60 min and being centrifuged at 1000 rpm for 3 min at room temperature, radioactivity was measured by MicroBeta^2^ (PerkinElmer). Agonistic and antagonistic activity were then evaluated in terms of the percentage of maximal response and percentage of inhibition against reference compounds (MOP: 300 nM DAMGO, DOP: 1000 nM DPDPE, KOP: 1000 nM U-69593, and NOP: 3000 nM nociceptin). EC_50_ and IC_50_ values were calculated by curve fitting using Prism 7 software. The assays were performed on two separate occasions, each in triplicate.

### 4.4. Animals

Male Crl:CD1 (ICR) mice (Japan CLEA, Tokyo, Japan) were used at 5 weeks of age for the pharmacokinetic analysis. Male C57BL/6J mice (Japan CLEA) were used at 7 weeks of age for the behavioral analyses. The mice were housed 4–6 per cage in an environment at 23 °C ± 1 °C and 50% ± 5% humidity with free access to food and water under a 12 h/12 h light/dark cycle. All experiments were performed with approval from the Institutional Animal Care and Use Committee at the Tokyo Metropolitan Institute of Medical Science.

### 4.5. Pharmacokinetic Analysis

After the oral administration of UD-030 once in male mice at doses of 0.5, 1, 2, and 4 mg/kg, the plasma concentration of UD-030 was measured. Blood samples (approximately 100 µL) were collected using heparinized micro-hematocrit capillary tubes (Kimble Chase, Vineland, NJ, USA) from the tail vein 0.5, 1, 2, 4, 8, and 24 h after drug administration (*n* = 3/group, collected 0.5, 2, and 8 h after administration; *n* = 3/group, collected 1, 4, and 24 h after administration). The blood samples were centrifuged at 15,000× *g* for 5 min at room temperature, and then the supernatant was collected as plasma and stored at −80 °C until analysis.

To measure UD-030 in plasma, 20 µL of the collected samples, 20 µL of the internal standard solution (5.00 ng/mL in acetonitrile), and 120 μL of acetonitrile were mixed, stirred, and centrifuged at 20,400× *g* for 5 min at 4 °C. Formic acid (0.1%, 80 μL) was then added to 80 μL of the supernatant, and the mixture was stirred and used as the sample for liquid chromatography–dual mass spectrometry (LC–MS/MS) using an API 4000 System (AB Sciex LLC, Framingham, MA, USA). Liquid chromatography was performed on a Cadenza CD-C18 column (3-µm particle size, 75 × 2.0 mm, Imtakt, Kyoto, Japan). The mobile phase was acetonitrile/0.1% formic acid (50/50) at a flow rate of 0.3 mL/min, and the injection volume was 20 µL. The following ion transitions were selected for quantitation: UD-030 (471.4 > 131.2 m/z) and internal standard (99.9% purified UD-030: 480.5 > 162.1 m/z). The amount of UD-030 in each sample was calculated by comparing the response of the analyte in the sample to a seven-point standard curve (0.1–100 ng/mL).

Using the peak area ratio of the analyte to the internal standard substance and the prepared concentration, the regression equation of the calibration curve (weighting of 1/x^2^) was determined, and the concentration was calculated. The preparation of calibration curves and calculation of the concentration were performed using Analyst 1.6.2 analytical software that was connected to the LC-MS/MS. The maximum plasma concentration (C_max_), time to reach the maximum concentration in plasma (T_max_), and area under the plasma concentration-time curve (AUC_0_–_24h_) were calculated based on the mean concentration at each blood collection time point using non-compartment analysis in Phoenix WinNonlin 6.1 software (Certara, Princeton, NJ, USA).

### 4.6. Conditioned Place Preference Test

The CPP test was performed using a two-compartment Plexiglas chamber. One compartment (175 mm width × 150 mm length × 175 mm height) had a black floor and walls with an equally spaced stainless-steel stripe-like grid on the floor. The other compartment had the same dimensions but had a white floor and walls with a stainless-steel grid on the floor. For the preconditioning and postconditioning phases, a T-style division with double 60 mm × 60 mm openings allowed the mice to access both compartments. During the conditioning phases, the openings were eliminated to restrict the mice to one of the compartments. The CPP apparatus was placed in a sound-attenuated, light-controlled box. On day 1 (preconditioning: habituation) and day 2 (preconditioning: pretest), the mice freely explored the two compartments for 900 s, and the time spent in each compartment during the exploratory period and locomotor activity were measured using an infrared detector (Neuroscience, Osaka, Japan). We selected a counterbalanced protocol to nullify each mouse’s initial preference as discussed previously [33]. Biased mice that spent more than 80% of the time (i.e., 720 s) on one side on day 2 or showed a difference of >200 s in the time spent in one side between days 1 and 2 were eliminated from subsequent procedures. Conditioning was conducted once daily for 4 consecutive days (days 3–6). The mice were injected with morphine (10.0 mg/kg, i.p.) or saline and immediately confined to the black or white compartment for 60 min on day 3. On day 4, the mice were alternately injected with saline or drug and immediately confined to the opposite compartment for 60 min. On days 5 and 6, the same conditioning as on days 3 and 4 was repeated. The assignment of the mice to the conditioned compartment was performed randomly and counterbalanced across subjects. During the postconditioning phase on day 7, the time spent in each compartment was measured for 900 s. The CPP score was designated as the time spent in the drug-paired compartment on day 7 minus the time spent in the same compartment in the preconditioning phase on day 2. The effects of UD-030 and NTX on the rewarding properties of morphine were tested in mice that orally received either UD-030 (3 and 10 mg/kg) or NTX (10 mg/kg) 60 min before each morphine injection (acquisition) or before the postconditioning phase on day 7 (expression).

The data were analyzed using two-way repeated-measures ANOVA followed by Sidak’s multiple-comparison *post hoc* test, one-way ANOVA followed by Dunnett’s multiple-comparison *post hoc* test, or Student’s *t*-test using GraphPad Prism software. Values of *p* < 0.05 were considered statistically significant.

## Figures and Tables

**Figure 1 ijms-24-03351-f001:**
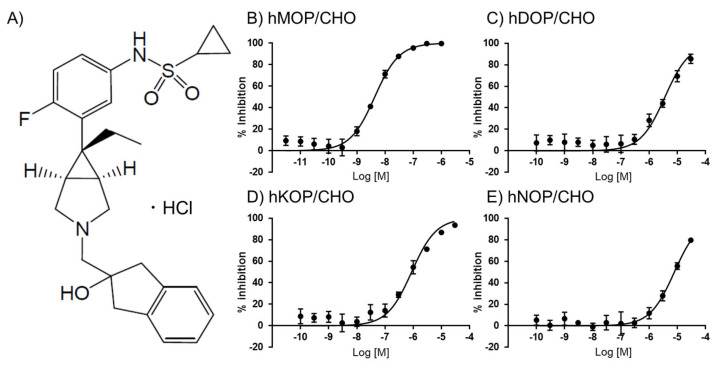
(**A**) Structure of UD-030. (**B**–**E**) Binding properties of UD-030 for displacement of the specific binding of 1 nM [^3^H]DAMGO, 0.7 nM [^3^H]DADLE, 1 nM [^3^H]U-69593, and 0.5 nM [^3^H]nociceptin to membranes of hMOR/CHO (**B**), hDOR/CHO (**C**), hKOR/CHO (**D**), and hNOP/CHO (**E**) cells, respectively. Assays were performed on two separate occasions, each in triplicate. The data are expressed as the mean ± SEM.

**Figure 2 ijms-24-03351-f002:**
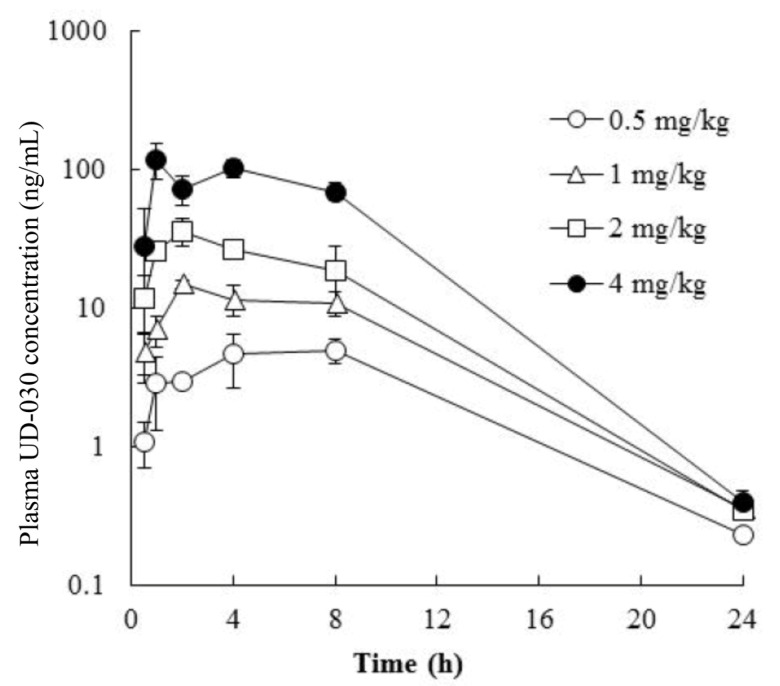
Plasma concentration vs. time profiles of UD-030. UD-030 was administered orally once in male mice over the dose range of 0.5 to 4 mg/kg. The data are expressed as the mean ± SD (*n* = 3).

**Figure 3 ijms-24-03351-f003:**
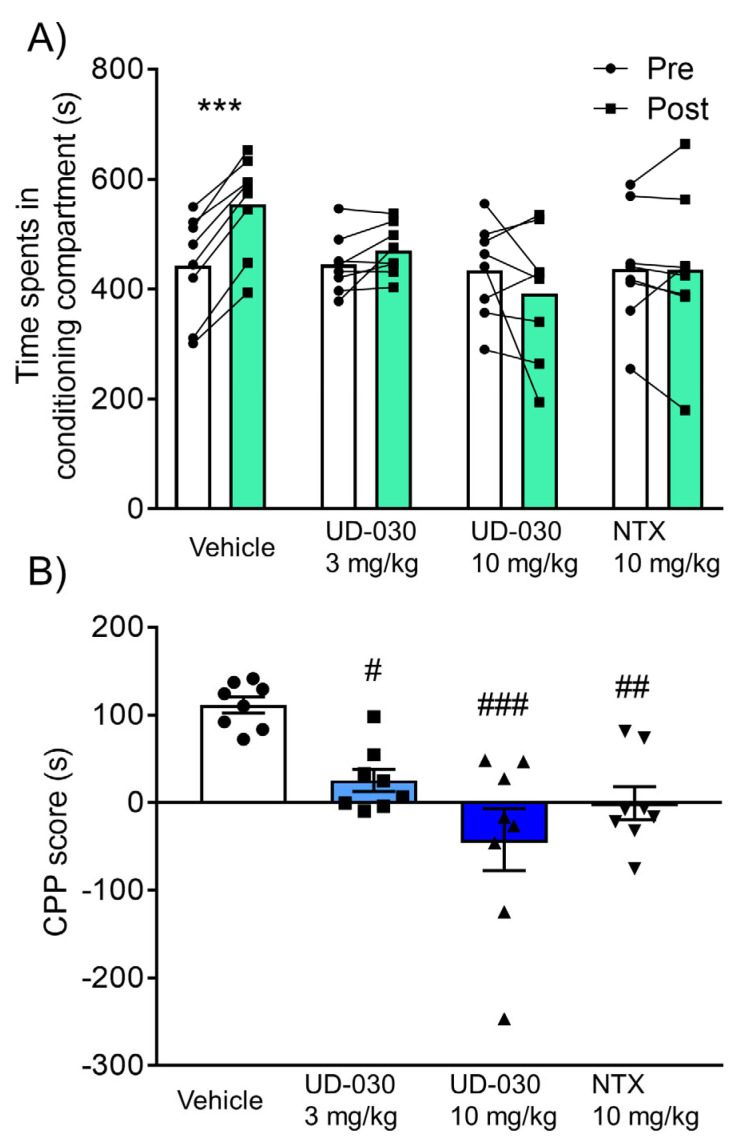
Inhibitory effect of UD-030 on the acquisition of morphine-induced conditioned place preference. (**A**) Time spent in the drug-paired compartment in the preconditioning phase (pre, white columns) and postconditioning phase (post, green columns). Mice (*n* = 8/group) were pretreated with UD-030 (0, 3, and 10 mg/kg, p.o.) or NTX (10 mg/kg, p.o.) 60 min before each morphine treatment (10 mg/kg, i.p.) during the conditioning phase. Lines that connect symbols represent values for individual mice. Columns represent the mean. *** *p* < 0.001, difference between pre- and postconditioning phases for each treatment. (**B**) Conditioned place preference (CPP) scores for each treatment of mice. The columns and vertical lines represent the mean ± SEM. *^#^ p* < 0.05, *^##^ p* < 0.01, *^###^ p* < 0.001, compared with vehicle-pretreated (control) mice.

**Figure 4 ijms-24-03351-f004:**
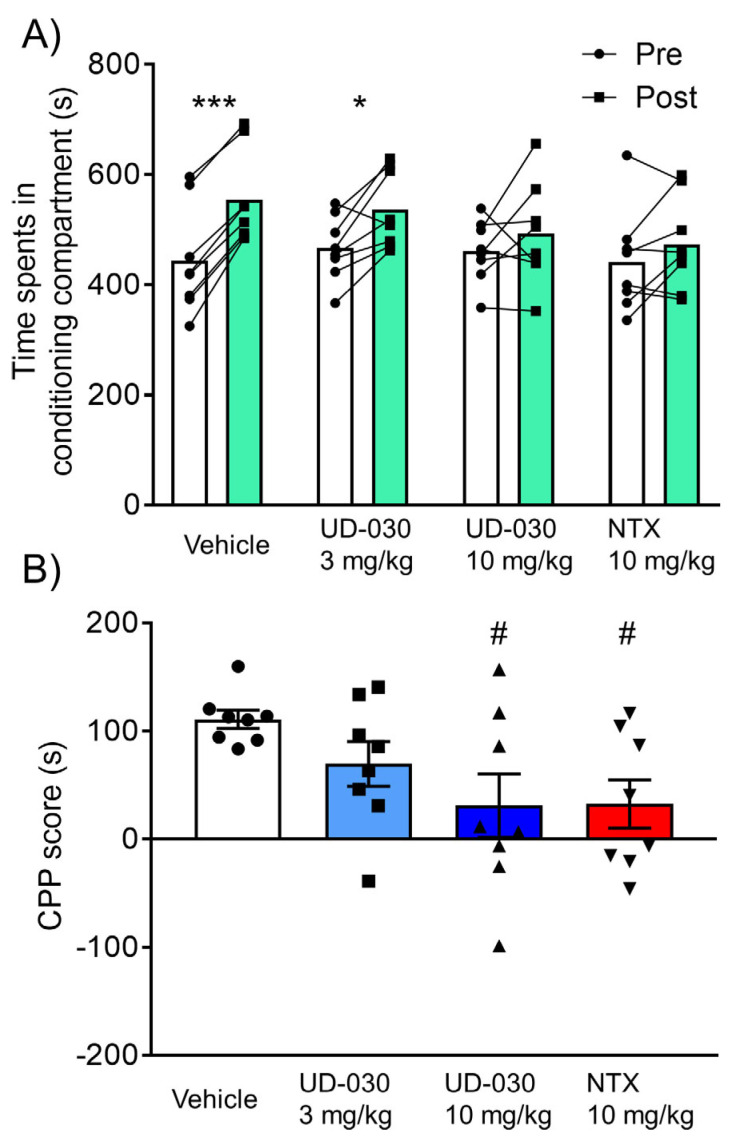
Inhibitory effect of UD-030 on the expression of morphine-induced conditioned place preference. (**A**) Time spent in the morphine (10 mg/kg, i.p.)-paired compartment in the preconditioning phase (pre, white columns) and postconditioning phase (post, green columns). Mice (*n* = 8/group) were pretreated with UD-030 (0, 3, and 10 mg/kg, p.o.) or NTX (10 mg/kg, p.o.) 60 min before the postconditioning phase. Lines that connect symbols represent the value of individual mice. Columns represent the mean. *** *p* < 0.001, * *p* < 0.05, difference between pre- and postconditioning phases for each treatment. (**B**) Conditioned place preference (CPP) scores for each treatment. The columns and vertical lines represent the mean ± SEM. ^#^
*p* < 0.05, compared with vehicle-pretreated (control) mice.

**Table 1 ijms-24-03351-t001:** In vitro actions of UD-030 and reference compounds against human opioid receptors and the NOP.

	hMOP/CHO	hDOP/CHO	hKOP/CHO	hNOP/CHO
Competitive binding assay				
UD-030 K_i_ (nM)	3.1	1800	460	1800
	[2.7–3.5]	[1500–2100]	[390–550]	[1600–2100]
Selectivity (vs. μ)		581	148	581
NTX K_i_ (nM) ^a^	0.23	38	0.25	ND
Selectivity (vs. μ)		165	1	ND
[^35^S]-GTPγS binding assay				
UD-030 EC_50_ (nM)	>10,000	>10,000	>10,000	>10,000
IC_50_ (nM)	1.7 [1.4–2.0]	850 [760–940]	85 [76–96]	330 [290–370]
DAMGO EC_50_ (nM)	2.5 [2.1–2.9]	—	—	—
DPDPE EC_50_ (nM)	—	1.8 [1.5–2.1]	—	—
U69593 EC_50_ (nM)	—	—	14 [13–15]	—
Nociceptin EC_50_ (nM)	—	—	—	16 [15–18]

^a^ K_i_ values for NTX were reported by Peng et al. (2007). —, not investigated in the present study. The data are expressed as means with [95% confidence interval (CI)].

**Table 2 ijms-24-03351-t002:** Plasma concentrations and pharmacokinetic parameters of UD-030 after a single oral administration.

Time (h)	Plasma Concentration (ng/mL)			
	0.5 mg/kg	1 mg/kg	2 mg/kg	4 mg/kg
0.5	1.09 ± 0.4	4.76 ± 1.9	11.8 ± 5.4	27.7 ± 24
1	2.86 ± 1.6	6.98 ± 1.7	25.5 ± 2.2	119 ± 35
2	2.96 ± 0.3	14.9 ± 0.9	36.1 ± 8.3	72.7 ± 18
4	4.61 ± 1.9	11.5 ± 2.9	26.7 ± 3.3	103 ± 15
8	4.95 ± 0.9	10.9 ± 2.1	18.8 ± 9.2	69.4 ± 10
24	0.231 ± 0	0.358 ± 0	0.35 ± 0	0.395 ± 0.1
T_max_ (h)	8.0	2.0	2.0	1.0
C_max_ (ng/mL)	4.95	14.9	36.1	119
AUC_0–24h_ (ng·h/mL)	72.3	176	350	1220

The data are expressed as the mean ± SD (*n* = 3).

## Data Availability

Publicly available datasets were analyzed in this study. These data can be found at https://drive.google.com/drive/folders/1ffjrs6CdVPdbsDg7p6lOi84OTZWTAYqL?usp=sharing (accessed on 22 December 2022).

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
