# Peer review of "Inhibitory Effects of a Novel μ-Opioid Receptor Nonpeptide Antagonist, UD-030, on Morphine-Induced Conditioned Place Preference"

_ijms, 2023, doi:10.3390/ijms24043351_

Round 1

Reviewer 1 Report

This is an interesting article which shows important results in clinical uses.

Conglatulation.

Author Response

Thank you very much for your high evaluation of our study.

Reviewer 2 Report

Thank you for the opportunity to review this interesting paper

Some important issues must be addressed

Introduction

Opioid prescriptions have increased annually, especially in 30 the United States - this is not a reliable description of the scope of the problem, please address 

Other regions also have similar issues and the underuse in other parts of the world is also concerning

Some moderately potent ligands are available, but few optimal nonpeptide antagonists have been developed for the MOP. - elaborate on the issues with peptides and reason for searching for nonpeptide ligands

It is mentioned here - Nonpeptide selective antagonists of the MOP could be potential therapeutic agents for addiction in terms of central transfer - but needs further clarification to support the need for the present study

All figures belong to the results section not introduction

Last paragraph in the introduction should be the study aim

methods

The biggest concern is the PK study, it must be further explained 

it shows  insufficient characterisation of the pharmacokinetic profiles with limited non compartmental analysis

Why weren’t other parameters determined? Why did authors opt just for oral administration without comparison to intravenous route?

Was in silico prediction of pk performed before moving to animal studies?

How were the time points for the study selected?

Why did authors include only 3 animals per group? This needs to be justified.

How was the dose to be administered selected?

Is the current animal model suitable for this type of Pk study?

What was used as internal standard for the analysis? Was the method validated? Are chromatograms available?

Discussion

In the present study, we found that the novel selective MOP nonpeptide antagonist UD-030 may be promising for the treatment of opioid use disorder - this claim is a bit over optimistic considering the preliminary nature of current findings based on a single in vivo study. Also, it is not mentioned that it is difficult to determine how these findings 

may translate into different physiological responses in vivo in humans

The pharmacology of the opioid receptors in insufficiently discussed i.e.  no mention in complex pathways or conformational dynamics is mentioned 

Limitation section is missing 

Author Response

Response to Reviewer 2

Some important issues must be addressed

Introduction

Opioid prescriptions have increased annually, especially in 30 the United States - this is not a reliable description of the scope of the problem, please address 

Other regions also have similar issues and the underuse in other parts of the world is also concerning

Response: According to the reviewer’s comments, we changed the following sentence in the Introduction:

From:

“Opioid prescriptions have increased annually, especially in the United States. As a result, opioid addiction and the opioid overdose epidemic (i.e., the "opioid crisis") have become increasingly more serious.”

To:

“However, the regular nonmedical use of opioids, prolonged use, misuse, and use without medical supervision can lead to opioid addiction and other health problems. Opioid addiction and the opioid overdose epidemic (i.e., the "opioid crisis") have become increasingly more serious worldwide, although there are countries where prescriptions are low and adequate pain management is not being provided [1].”

Some moderately potent ligands are available, but few optimal nonpeptide antagonists have been developed for the MOP. - elaborate on the issues with peptides and reason for searching for nonpeptide ligands

It is mentioned here - Nonpeptide selective antagonists of the MOP could be potential therapeutic agents for addiction in terms of central transfer - but needs further clarification to support the need for the present study

Response: According to the reviewer’s comments, we changed the following sentence in the Introduction:

From:

“Nonpeptide selective antagonists of the MOP could be potential therapeutic agents for ad-diction in terms of central transfer.”

To:

“Peptides generally cannot be administered orally because they are quickly degraded by digestive enzymes in the gastrointestinal tract. Additionally, in many cases, long-chain peptides have low membrane permeability because of the high number of water-soluble functional groups, resulting in poor pharmacokinetics in the body. Although peptide preparations that mitigate these disadvantages have been developed, there have been few successful examples of their medical use. Thus, nonpeptide selective MOP antagonists could be potential oral therapeutic agents for addiction.”

All figures belong to the results section not introduction

Response: According to the reviewer’s comments, we changed the locations of the figure citations.

Last paragraph in the introduction should be the study aim

Response: According to the reviewer’s comments, we changed the following sentence in the Introduction:

From:

“The present study biologically and pharmacologically evaluated a novel nonpeptide ligand, UD-030, as a selective MOP antagonist. In vitro studies showed that UD-030 is a highly selective antagonist of the MOP over the DOP, KOP, and NOP. The results of behavioral pharmacology studies indicated that UD-030 suppresses the acquisition and expression of morphine-induced conditioned place preference (CPP), and its effects were comparable to the clinically used NTX.”

To:

“The aim of the present study was to determine the binding selectivity of UD-030, a novel nonpeptide ligand, for each opioid receptor subtype and its agonist/antagonist activity. Conditioned place preference (CPP) tests were conducted to test the inhibitory effect of UD-030 on morphine preference and compare it to the clinically used NTX.”

Methods

The biggest concern is the PK study, it must be further explained 

it shows  insufficient characterisation of the pharmacokinetic profiles with limited non compartmental analysis

Why weren’t other parameters determined? Why did authors opt just for oral administration without comparison to intravenous route?

Was in silico prediction of pk performed before moving to animal studies?

How were the time points for the study selected?

Why did authors include only 3 animals per group? This needs to be justified.

How was the dose to be administered selected?

Is the current animal model suitable for this type of Pk study?

What was used as internal standard for the analysis? Was the method validated? Are chromatograms available?

Response: The PK study was conducted with a small number of mice by selecting a general time course and method to preliminarily confirm the approximate timing of peak plasma concentration and half-life before conducting the behavioral experiments for the purpose of developing the drug for oral administration. Thus, this was insufficient to guarantee reliability and reproducibility. The following sentences were added in the Results and Discussion:

Results:

“To investigate the potential usefulness of the nonpeptide ligand UD-030 as an oral agent, we conducted a preliminary validation in a small number of mice to determine whether it is absorbed after oral administration and predict the peak and duration of its effects after administration.”

Discussion:

“One of the main limitations of the present study, in addition to not performing conditioning with UD-030 alone, was the lack of assurance of the reliability and reproducibility of pharmacokinetic studies. The present study was conducted by selecting a general time course and method for a preliminary positioning study to confirm the approximate timing of peak plasma concentration and half-life before conducting the behavioral experiments for the purpose of developing the drug for oral administration. Future validation with matching animal species and doses and comparisons with other routes of administration will be required.”

If the editor and reviewer deems it necessary, then we could present Fig. 2 and Table 2 as supplementary items because the PK study was not the main focus of the present study.

Discussion

In the present study, we found that the novel selective MOP nonpeptide antagonist UD-030 may be promising for the treatment of opioid use disorder - this claim is a bit over optimistic considering the preliminary nature of current findings based on a single in vivo study. Also, it is not mentioned that it is difficult to determine how these findings may translate into different physiological responses in vivo in humans

Response: According to the reviewer’s comments, we changed the following sentence in the Discussion:

From:

“In the present study, we found that the novel selective MOP nonpeptide antagonist UD-030 may be promising for the treatment of opioid use disorder.”

To:

“In the present study, we found that the novel selective MOP nonpeptide antagonist UD-030 may have the potential to be a seed compound for the treatment of opioid use disorder.”

We also mentioned additional limitations in the Discussion:

“Another limitation was the lack of reward-related experiments beyond CPP, such as self-administration and intracranial self-stimulation. The CPP test can be influenced by memory. Other reward-related studies should be conducted in the future. Moreover, further studies of UD-030 and its derivatives are required to verify various other physiological effects that are applicable to human studies. Putative agents, such as MOP antagonists, that reduce the nonmedical use of prescription opioids may also negatively impact patient compliance if aversive effects of the putative agent are too strong. Another consideration is the complementarity of pharmacokinetic profiles of paired drugs in a formulation.”

The pharmacology of the opioid receptors in insufficiently discussed i.e.  no mention in complex pathways or conformational dynamics is mentioned 

Response: Because the ligand we are developing is an antagonist, we think it is a bit off-topic to discuss much about the intracellular signaling pathway diversity of opioid receptors. According to the reviewer’s comments, we added the following sentences in the Discussion:

“The concept of 'functional selectivity' or 'biased agonism' has been attracting attention in recent years with regard to separating analgesic effects of MOP agonists from their adverse effects [29]. The activation of MOPs by agonists triggers various processes via downstream signaling through G proteins and β-arrestins. Although most MOP agonists that have been reported to date are non-biased ligands, G-protein signaling pathway-biased MOP ligands have been hypothesized to provide analgesia with fewer adverse effects [29]. Antagonistic effects of UD-030 on these biased and non-biased ligands will need to be tested in the future.”

Limitation section is missing 

Response: According to the reviewer’s comments, we mentioned several limitations in the Discussion:

“One of the main limitations of the present study, in addition to not performing conditioning with UD-030 alone, was the lack of assurance of the reliability and reproducibility of pharmacokinetic studies. The present study was conducted by selecting a general time course and method for a preliminary positioning study to confirm the approximate timing of peak plasma concentration and half-life before conducting the behavioral experiments for the purpose of developing the drug for oral administration. Future validation with matching animal species and doses and comparisons with other routes of administration will be required. Another limitation was the lack of reward-related experiments beyond CPP, such as self-administration and intracranial self-stimulation. The CPP test can be influenced by memory. Other reward-related studies should be conducted in the future. Moreover, further studies of UD-030 and its derivatives are required to verify various other physiological effects that are applicable to human studies. Putative agents, such as MOP antagonists, that reduce the nonmedical use of prescription opioids may also negatively impact patient compliance if aversive effects of the putative agent are too strong. Another consideration is the complementarity of pharmacokinetic profiles of paired drugs in a formulation.”

Reviewer 3 Report

This is an interesting and well-written submission on a novel compound with high selectivity as an antagonist for the mu-opioid receptor.    The experiments are described well and represent an initial effort to characterize the compound, using naloxone as a comparator.  The compound blocks the acquisition and expression of morphine in the conditioned place preference method, has good PK and could be a viable drug candidate.  One concern is whether there is sufficient justification provided to warrant the possible introduction of a compound that, despite its specificity, is very similar in its pharmacology, presented here, to naloxone.  Emphasizing possible merits of this compound would be helpful.

Three questions/concerns, though relatively trivial, could be mentioned for the authors to consider.  1. It might be useful to include the species in the abstract.  2.  On line 223, I believe the sentence should read '... are currently prescribed '.  Finally, this sentence might be clarified:  "Thus, opioid analgesics are thought to exert their analgesic effects through both the MOP and KOP".  There has been a long-standing view by some that both MOP and KOP contribute to the analgesic effects of drugs such as oxycodone but, with the exception of buprenorphine perhaps, the effects of most analgesics are clearly mediated through the mu-receptor.  Some KOP compounds do produce analgesic effects, but the statement might be clarified. 

Author Response

Response to Reviewer 3

This is an interesting and well-written submission on a novel compound with high selectivity as an antagonist for the mu-opioid receptor. The experiments are described well and represent an initial effort to characterize the compound, using naloxone as a comparator.  The compound blocks the acquisition and expression of morphine in the conditioned place preference method, has good PK and could be a viable drug candidate.  One concern is whether there is sufficient justification provided to warrant the possible introduction of a compound that, despite its specificity, is very similar in its pharmacology, presented here, to naloxone. Emphasizing possible merits of this compound would be helpful.

Response: According to the reviewer’s comments, we added the following sentences in the Discussion:

“In the present study, the novel nonpeptide ligand UD-030 showed high affinity for the MOP and higher selectivity for the MOP compared with NTX. This may indicate that UD-030 has the potential to be a centrally transferable drug with fewer side effects that are caused by low subtype selectivity.”

Three questions/concerns, though relatively trivial, could be mentioned for the authors to consider.  1. It might be useful to include the species in the abstract. 

Response: According to the reviewer’s comments, we added the mouse species in the Abstract.

  1. On line 223, I believe the sentence should read '... are currently prescribed '.

Response: This was corrected.

Finally, this sentence might be clarified:  "Thus, opioid analgesics are thought to exert their analgesic effects through both the MOP and KOP".  There has been a long-standing view by some that both MOP and KOP contribute to the analgesic effects of drugs such as oxycodone but, with the exception of buprenorphine perhaps, the effects of most analgesics are clearly mediated through the mu-receptor.  Some KOP compounds do produce analgesic effects, but the statement might be clarified. 

Response: According to the reviewer’s comments, we changed the following sentence in the Discussion:

From:

“Thus, opioid analgesics are thought to exert their analgesic effects through both the MOP and KOP and produce dependence primarily through the MOP.”

To:

“Our previous studies with inbred and MOP gene knockout mice have also shown that the MOP plays a major role in analgesic and rewarding effects of opioids [5,6], and the KOP is also partially involved in their analgesic effects on visceral pain [7,8]. Thus, opioid analgesics are thought to exert their analgesic effects primarily through the MOP and partially through the KOP, but they produce dependence primarily through the MOP.”

Reviewer 4 Report

The Ide et al. manuscript is aiming to prove that UD-030 is a selective µ-opioid receptor antagonist and can be used to treat opioid-use disorders. To measure the affinity of UD-030 for µ-opioid receptor and other opioid receptors, the authors used radioligand binding assay, [35s]-GTPgammaS binding assay, and claim that UD-030 is a selective antagonist for µ-opioid receptor. Additionally, the authors measured the pharmacokinetic properties of UD-030 in mice after oral administration. Finally, the author examined the effect of UD-030 on morphine-induced CPP. The misuse of opioids is a global problem, especially in the US, where people are fighting against both opioids and chronic pain epidemics. A selective antagonist of the µ-opioid receptor, the main target of most commonly used opioids in the clinic, would benefit the whole society. Thus, this is a meaningful study. However, more experiments are needed to convince readers that UD-030 is a selective µ-receptor antagonist. Below are my major and minor concerns about this manuscript: Major concerns: 1. As the authors mentioned in the introduction, opioids are widely used for pain management. The misuse of prescription and non-prescription opioids lead to the opioid epidemic in the US. Thus, to increase the impact of UD-030 as a selective µ-opioid receptor, the authors should examine whether UD-030 could reverse the analgesic effect of opioids. Most pain behaviors are relatively easy to test. 2. Fig.3 indicates that high-dose (10 mg/kg) of UD-030 induces a place aversion in the mice. If UD-030 is aversive, it will likely affect the CPP results. Currently, the authors interpret the reversed morphine-induced CPP after UD-030 administration as the consequence of the antagonistic effect of UD-030 on µ-opioid receptors. How do the authors exclude the possibility that the CPP results are due to the aversive property of UD-030? Whether UD-030 injection alone causes place aversion? 3. Related to major concern 2, is there any side effect the authors observed after UD-030 administration for mice? The authors should mention it somewhere in the manuscript because the side effects may change mouse behavior during CPP. Minor concerns: 1. English needs to be further polished. 2. For the radioligand binding assay, the final concentrations of MOP, DOP, KOP, and NOP are different (lines 254 and 255). However, the author did not explain it or cite other studies to support it. 3. In table 1, the EC50 value of DAMGO in the [35s]-GTPgammas binding assay is dramatically different from other studies (https://www.ncbi.nlm.nih.gov/pmc/articles/PMC1573101/). What could be the reason? 4. P.O. (line 124), the authors should write the full name in the first instance. 5. Two-way ANOVA with post-hoc pairwise comparisons would be better than one-way ANOVA for Fig.3A and Fig.4A.

Please see the comments in the attached file.

Author Response

Response to Reviewer 4

The Ide et al. manuscript is aiming to prove that UD-030 is a selective µ-opioid receptor antagonist and can be used to treat opioid-use disorders. To measure the affinity of UD-030 for µ-opioid receptor and other opioid receptors, the authors used radioligand binding assay, [35s]-GTPgammaS binding assay, and claim that UD-030 is a selective antagonist for µ-opioid receptor. Additionally, the authors measured the pharmacokinetic properties of UD-030 in mice after oral administration. Finally, the author examined the effect of UD-030 on morphine-induced CPP. The misuse of opioids is a global problem, especially in the US, where people are fighting against both opioids and chronic pain epidemics. A selective antagonist of the µ-opioid receptor, the main target of most commonly used opioids in the clinic, would benefit the whole society. Thus, this is a meaningful study. However, more experiments are needed to convince readers that UD-030 is a selective µ-receptor antagonist. Below are my major and minor concerns about this manuscript:

Major concerns: 1. As the authors mentioned in the introduction, opioids are widely used for pain management. The misuse of prescription and non-prescription opioids lead to the opioid epidemic in the US. Thus, to increase the impact of UD-030 as a selective µ-opioid receptor, the authors should examine whether UD-030 could reverse the analgesic effect of opioids. Most pain behaviors are relatively easy to test.

Response: As the reviewer states, the assessment of some pain-related behaviors is indeed easily possible. However, the signaling systems and nervous system tracts that are involved differ according to the type of pain, and analgesic effects that are exerted by each opioid analgesic are likewise different. Because UD-030 is thought to inhibit opioid analgesia as well as other antagonists with binding affinity for the MOP, a fixed-dose combination with an agonist would be realistic for future clinical applications. Such studies are future challenges. According to the reviewer’s suggestion, we revised the following sentences in the Discussion:

From:

“At present, it is not possible to separate the analgesic effect from dependence via MOPs. While the problem of opioid use disorder continues to grow, the fact that patients with opioid use disorder still suffer from chronic pain is also a major problem from a treatment perspective [27,28]. Unclear is whether opioid receptor agonist treatment should be maintained or tapered for patients with both chronic pain and opioid use disorder. For these patients, methadone or a buprenorphine/naloxone fixed-dose combination is currently prescribed as opioid replacement therapy. Fixed-dose combinations of MOP-selective antagonists and opioid receptor agonists may lead to new therapies that partially suppress MOPs while maintaining analgesic effects of KOP activation. Although not directly tested in the present study, selective MOP antagonists may be developed for the treatment of chronic pain and comorbid addiction with fewer side effects.”

To:

“At present, it is not possible to separate the analgesic effect from dependence via MOPs. Although not studied here, UD-030, like other opioid receptor antagonists with binding affinity for MOP, could be thought to inhibit analgesic effects of opioids. While the problem of opioid use disorder continues to grow, the fact that patients with opioid use disorder still suffer from chronic pain is also a major problem from a treatment perspective [31,32]. Unclear is whether opioid receptor agonist treatment should be maintained or tapered for patients with both chronic pain and opioid use disorder. For these patients, methadone or a buprenorphine/naloxone fixed-dose combination are currently prescribed as opioid replacement therapy. Fixed-dose combinations of MOP-selective antagonists and opioid receptor agonists may lead to new therapies that partially suppress MOPs while maintaining analgesic effects of KOP activation. Future studies should determine specific ratios of combinations that can maintain analgesic effects while reducing dependence potential such that selective MOP antagonists are developed for the treatment of chronic pain and comorbid addiction with fewer side effects.”

  1. Fig.3 indicates that high-dose (10 mg/kg) of UD-030 induces a place aversion in the mice. If UD-030 is aversive, it will likely affect the CPP results. Currently, the authors interpret the reversed morphine-induced CPP after UD-030 administration as the consequence of the antagonistic effect of UD-030 on µ-opioid receptors. How do the authors exclude the possibility that the CPP results are due to the aversive property of UD-030? Whether UD-030 injection alone causes place aversion?

Response: As indicated by the reviewer, we did not test CPA with UD-030 alone, which is a limitation. Nonetheless, antagonism of the expression of morphine-induced CPP suggests that this is not simply counteracting effects of CPP and CPA. We added the following sentences in the Discussion:

“Not only NTX but also other opioid receptor antagonists, such as naloxone, have been reported to cause conditioned place aversion (CPA) [15,16]. No conditioning with UD-030 treatment alone was performed in the present study, which is a limitation. However, a slight, although nonsignificant, tendency toward CPA was observed in the 10 mg/kg UD-030 and morphine conditioning group, suggesting that UD-030, like other opioid antagonists, could produce CPA. Nonetheless, UD-030 inhibited the expression of morphine-induced CPP after acquisition, indicating that UD-030-induced CPA may not simply counteract morphine-induced CPP but may suppress rewarding effects of morphine via the modulation of endogenous opioid neural activity.”

  1. Related to major concern 2, is there any side effect the authors observed after UD-030 administration for mice? The authors should mention it somewhere in the manuscript because the side effects may change mouse behavior during CPP.

Response: According to the reviewer’s comments, we added the following sentences in the Results:

“In the postconditioning phase, there were no significant differences in the number of transitions between compartments among groups (one-way ANOVA: F3,28 = 0.82, p ­= 0.49), indicating that 10 mg/kg UD-030 and NTX did not exert sedative effects at this dose. No other notable behavioral abnormalities were observed in mice after high-dose (10 mg/kg, p.o.) UD-030 treatment during the session.”

Minor concerns:

  1. English needs to be further polished.

Response: The revised manuscript was thoroughly proofread by a native English-speaking editor.

  1. For the radioligand binding assay, the final concentrations of MOP, DOP, KOP, and NOP are different (lines 254 and 255). However, the author did not explain it or cite other studies to support it.

Response: The final concentration that is mentioned in the Methods was the concentration that was diluted as a result of adding Polytron homogenizer washing solution, etc., and is not the concentration that was intentionally adjusted. This method is not specifically described, but we believe it is a generally accepted method that can be found in other reports (Naunyn-Schmiedeberg’s Arch Pharmacol [2000] 361:498).

  1. In table 1, the EC50 value of DAMGO in the [35s]-GTPgammas binding assay is dramatically different from other studies (https://www.ncbi.nlm.nih.gov/pmc/articles/PMC1573101/). What could be the reason?

Response: In the paper mentioned by the reviewer, the authors stated, “SH-SY5Y cells and C6 rat glioma cells stably transfected with a rat mu (C6(μ)) or delta (C6(δ)) opioid receptor were used.” The expression cells and type of receptor (human vs. rat) that were used in this study were different from each other. The EC50 values may deviate from one experimental system to another, and it is important to compare EC50 values with a positive control that is tested at the same time. In fact, there is no significant deviation from the values in another paper (https://pubmed.ncbi.nlm.nih.gov/21550962/).

  1. O. (line 124), the authors should write the full name in the first instance.

Response: This modification was made as indicated.

  1. Two-way ANOVA with post-hoc pairwise comparisons would be better than one-way ANOVA for Fig.3A and Fig.4A.

Response: According to the reviewer’s comments, we revised our statistical methods and added or changed relevant sentences in the Results and Methods.

Round 2

Reviewer 2 Report

The authors have adressed all comments

Reviewer 4 Report

The authors solved all of my concerns, the manuscript could be published as it is now.